# Identification and Expression Analysis of *DFR* Gene Family in *Brassica napus* L.

**DOI:** 10.3390/plants12132583

**Published:** 2023-07-07

**Authors:** Xingzhi Qian, Wenyin Zheng, Jian Hu, Jinxu Ma, Mengyuan Sun, Yong Li, Nian Liu, Tianhua Chen, Meiqi Wang, Ling Wang, Xinzhe Hou, Qingao Cai, Zhaoshun Ye, Fugui Zhang, Zonghe Zhu

**Affiliations:** College of Agronomy, Anhui Agricultural University, Hefei 230036, China; qxz799252@163.com (X.Q.); zhengwenyin_75@163.com (W.Z.); 21720163@stu.ahau.edu.cn (J.H.); 15911922305@163.com (J.M.); m997295123@163.com (M.S.); liyong1313@163.com (Y.L.); 15851859033@163.com (N.L.); 13830547235@163.com (T.C.); w1891924@163.com (M.W.); ling05502023@163.com (L.W.); houxinzhe9812@163.com (X.H.); cqa13937384025@163.com (Q.C.); yezhaoshun@stu.ahau.edu.cn (Z.Y.); zhfugui91@163.com (F.Z.)

**Keywords:** *Brassica napus* L., *DFR* gene family, gene expression, bioinformatics analysis

## Abstract

Dihydroflavonol 4-reductase (*DFR*) is a key enzyme in the flavonoid biosynthetic pathway and is essential for the formation of plants’ color. In this study, 26 *BnDFR* genes were identified using 6 *Arabidopsis DFR* genes as reference. The physicochemical properties, subcellular localization, and conserved structure of *BnDFR* proteins were analyzed; the evolutionary relationship, collinearity analysis, and expression characteristics of *BnDFR* genes were studied; and the correlation between the expression level of *BnDFR* genes and anthocyanin content in rape petals were analyzed. The results showed that the 26 *BnDFRs* were located in chloroplasts, cytoplasm, nuclei, and mitochondria, distributed on 17 chromosomes, and divided into 4 groups; members of the same group have a similar function, which may be related to the environmental response elements and plant hormone response elements. Intraspecific collinearity analysis showed 51 pairs of collinear genes, and interspecific collinearity analysis showed 30 pairs of collinear genes. Analysis of the expression levels of *BnDFRs* and anthocyanin content in different color rape petals showed that *BnDFR6* and *BnDFR26* might play an important role in the synthesis of anthocyanins in rape petals. This provides theoretical guidance for further analysis of the anthocyanin anabolism mechanism involved in the *DFR* gene in *Brassica napus*.

## 1. Introduction

*Brassica napus* L. (AACC, 2n = 38) is one of the most important oil crops in the world, which is widely grown as the first oil source of edible vegetable oil made in China [1]. *B. napus* originated in the Mediterranean about 7500 years ago and is an allopolyploid species derived from an interspecific cross between two diploid ancestors, *B. rapa* (AA, 2n = 20) and *B. oleracea* (CC, 2n = 18). Therefore, there are two sets of subgenomes in *B. napus*, namely the A subgenome and the C subgenome. In the world, with the continuous improvement in people’s living quality and the rapid development of the recent picnic industry, rape is not only an important source of edible vegetable oil, vegetables, and fodder [2], but also has a high ornamental value. Rape tourism also gradually began to rise, and each region, according to its unique geographical characteristics and cultural advantages, actively developed leisure agriculture and rural tourism in order to create colorful rape flower sea tourism forms [3,4]. The color of rape is the key character that determines its ornamental value. Therefore, it is of great significance to cultivate new varieties of rape with rich colors to promote the multifunctional development and utilization of rape.

Understanding the molecular mechanism of color formation in rape can greatly promote the process of variety cultivation of colorful rape. In recent years, research on the molecular mechanism of flower color and gene mapping has been gradually carried out [5,6]. Zhao et al. found that the flower color of ornamental plants could be affected by *CHS* (chalcone synthase), *CHI* (chalcone isomerase), *F3H* (flavanone-3-ans), *DFR*, *ANS*, and *UFGT* genes associated with anthocyanin metabolism [7]. Zhang et al. mapped the dominant gene of white flowers within 361 kb on chromosome C03 by fine mapping and obtained the candidate gene *BnaC03.CDD4* by homologous cloning. *BnaC03.CCD4* may use δ-carotene and/or α-carotene as its substrates and break the double bond at positions 9 and 10 preferentially, leading to the release of volatile alpha-ionic ketone and the relatively low accumulation of violet xanthin and other carotenoids, turning yellow flowers into white ones [8]. Additionally, the study found that the rape color is formed by chlorophyll, carotenoid, anthocyanin, and alkaloid [9]. In particular, anthocyanins not only determine the color of rape but also play an important role in the protection of the plant itself and the attraction of pollinators. Anthocyanins are widely used in the prevention and treatment of various diseases due to their antioxidant, anti-inflammatory, and anti-aging properties [10] and have attracted wide attention worldwide. Therefore, it is of great significance to study the genes regulating anthocyanin synthesis and further analyze their molecular mechanism for the color breeding of *B. napus*.

The dihydroflavonol 4-reductase (*DFR*) family is a subfamily of the NAD(P)-dependent epimerase/dehydratase family involved in the biosynthesis of anthocyanins, proanthocyanidins, and other flavonoids, widely found in plants, is an important regulatory enzyme that controls the rate of anthocyanin biosynthesis by controlling the direction of carbon flux in the anthocyanin biosynthesis pathway [11]. It has been shown that *DFR* is associated with anthocyanin accumulation in several horticultural plants (such as kale) [12]. *DFR* catalyzes dihydrokaempferol, dihydroquercetin, and dihydromyricetin to produce corresponding colorless anthocyanins: leucopelargoridin, leucodelphinidin, and leucodelphinidin, identified as key regulatory sites in anthocyanidin biosynthesis [13]. The *DFR* gene plays an indispensable role in the process of ornamental plant color formation and has been systematically reported in *Arabidopsis* [14,15,16,17,18,19], rice [18,19], *Ginkgo* [13], and other plants. The *DFR* gene was first isolated from corn and goldfish by O ‘Reilly et al. [20] using the translocon labeling method and later cloned from *Petunia* [21] and other species (*Gerbera* [22], *Ginkgo* [13], *Scutellaria* [23], etc.). However, the mechanism of *DFR* in the formation of rape color is still unclear. Genome-wide identification can identify genes with the same domain, which is the functional unit of a protein, and having the same domain often has the same function. Therefore, color-specific genes in rape can be identified by reference to the genes related to color that have been studied.

In order to clarify whether the *DFR* gene has a regulatory effect on the color formation of rape, we comprehensively analyzed the members of the *DFR* gene family in *B. napus*, and a bioinformatics analysis was conducted on the physicochemical properties, subcellular localization, and conserved structure of 26 *BnDFR* proteins. Then, the evolutionary relationship, collinearity analysis, and expression characteristics of *BnDFR* genes were studied. Lastly, we analyzed the relationship between the expression level of the *DFR* genes and the anthocyanin content in flowers, which laid a theoretical foundation for clarifying the mechanism of action of the *DFR* gene involved in the regulation of flower color in *B. napus* and provided an important basis for the analysis of the biological function of the *DFR* gene.

## 2. Results

### 2.1. Analysis of BnDFR Family Proteins

To understand the evolution of the *DFR* family of *B. napus*, blast analysis is performed using the *DFR* proteins sequence from *Arabidopsis thaliana*; as a result, a total of 26 *BnDFR* genes are selected with an E-value < 1 × 10^−50^ in the AC genome, among which there are 14 *DFR* genes in the A subgenome and 12 in the C subgenome, designated as *BnDFR1*–*BnDFR26* (Appendix A). The analysis of the physicochemical properties and subcellular localization of *DFR* genes showed that proteins are significantly different (Table 1). The number of amino acids is 223~385, the molecular weight (MW) is 25.08~42.93 kD, and the theoretical isoelectric point (pI) is 4.93~8.79. In addition to *BnDFR5* and *BnDFR11*, all the *DFR* proteins are acidic proteins. In terms of protein stability, *BnDFR5* and *BnDFR18* are unstable proteins, and all the others are stable proteins. The grand average of hydrophobicity (GRAVY) of *DFR* family members is less than 0, indicating that all *DFR* proteins are hydrophilic proteins. Subcellular localization results showed that *DFR* proteins are distributed in the nucleus, cytoplasm, chloroplast, and mitochondria. According to the 3D structure prediction website PHYRE2 (http://www.sbg.bio.ic.ac.uk/phyre2/html (accessed on 4 April 2023)) [24], the 3D structure of 26 *BnDFR* proteins are obtained (Figure 1), and the secondary structure information of the 26 proteins is analyzed by SOPMA (https://npsa-prabi.ibcp.fr (accessed on 4 April 2023)) [25]. The proportion of α-helices in the family is 39.71% (*BnDFR4*) to 48.19% (*BnDFR6*), and the proportion of random coils in the family ranges from 29.35% (*BnDFR6*) to 41.28% (*BnDFR11*). Except for α-helices and random coils, there are extended strands and β-turns. The proportion of extended strands is 11.95% (*BnDFR10*)–17.03% (*BnDFR6*), and the proportion of β-turns is 4.93% (*BnDFR5*)–7.89% (*BnDFR18*). This indicates that *BnDFR* proteins are mainly composed of α-helices and random coils (Table 2).

### 2.2. Chromosome Localization and Collinearity Analysis of DFR Family

Based on the chromosome annotation information of *B. napus*, the identified *BnDFR* genes are mapped (Figure 2). According to the results of gene distribution, 26 *BnDFR* genes are unequally distributed on 17 chromosomes (9 in the A subgenome and 8 in the C subgenome), and only A04 and C07 are not distributed. In order to explore the evolutionary process of the *BnDFR* gene family, the intraspecies collinearity (Figure 3A) and interspecies collinearity (Figure 3B) are analyzed. Intraspecies collinearity analysis detected 51 pairs of collinear genes in 20 *BnDFR* genes, among which the A07 chromosome had the most collinear genes (16 pairs of collinear genes), indicating that gene replication greatly promoted the amplification of the *DFR* gene in the *B. napus* genome. Interspecies collinearity analysis showed that there are a large number of directly homologous *DFR* genes between *B. napus* and *A. thaliana*, and 21 *BnDFR* genes showed 30 pairs of collinearity with *A. thaliana*, and each *AtDFR* gene had 2 or more homologous genes in *B. napus*. The results indicated that *BnDFR* genes are significantly amplified during gene replication and brassica polyploidy events. In conclusion, most of the *BnDFR* genes remain intact during evolution.

### 2.3. Evolutionary Analysis of DFR Gene Family

In order to further reveal the evolutionary relationship of the *DFR* gene family, the gene structure, conserved motif, and conserved domain are predicted and divided into four groups based on the phylogenetic tree (Figure 4). According to the phylogenetic tree, the relationship between the members of the *DFR* gene family can be seen more directly. There are four copies of *AtDFR1*, four copies of *AtDFR2*, three copies of *AtDFR3*, four copies of *AtDFR4*, four copies of *AtDFR5*, and two copies of *AtDFR6* in *B. napus*. The five genes, *BnDFR5*, *BnDFR11*, *BnDFR12*, *BnDFR21*, and *BnDFR22*, are homologous to *AtDFR3*, but their genetic relationship has a distance. It is possible that there are great changes during the formation and evolution of *B. napus*.

To understand the structural characteristics of *DFR* genes, the TBtools software is used to analyze the gene structure and intron/exon arrangement of DFRs (Figure 5). The results showed that the number of introns of all DFRs is between three and six, but it varies less within the same group. There are three to six introns in Group I, five introns in Group II, three to five introns in Group III, and five introns in Group IV. The MEME online analysis website and TBtools software are used to analyze the conserved motif of 32 *DFR* proteins (Figure 5). The results showed that DFRs retained a lot of conserved motifs in the evolutionary process. The figure shows the distribution of 20 highly conserved motifs in 32 *DFR* protein sequences. It is obvious that the conserved motifs of 32 *DFR* proteins mainly include motif 1, motif 11, motif 2, motif 8, motif 7, motif 5, motif 3, motif 10, motif 4, motif 9, and motif 6, and the types and amounts of conserved motifs are similar among *DFR* within the same group. Group I includes motifs 19 and 12; Group II includes motif 15 and motif 18; Group III includes motif 12 and motif 15; and Group IV includes motif 20, motif 17, and motif 13, in which conserved motifs of some *DFR* proteins are lost, and different conserved motifs may reveal the causes of functional differentiation of different groups. Therefore, we studied the conserved functional domains based on conserved motifs. Domain refers to the independent stable structural region composed of different secondary structures and super secondary structures in proteins, which is also the functional unit of proteins; different domains are often associated with different functions. The FR_SDR_e domain of Group I and Group II indicated that motif 19, motif 12, and motif 15 do not change the function of the *DFR* protein. The domain of *BnDFR5* and *BnDFR11* is the NADB_Rossmann superfamily, which is due to the loss of part of the motif, leading to functional changes. The domain of Group III is the NADB_Rossmann superfamily, this result indicated that motif 18 insertion caused changes in protein function, and PLN02214 is a member of the NADB_Rossmann superfamily. Motif 20 and motif 17 are inserted in Group IV, which caused changes in protein function, in which PLN00198 is the only member of the PLN00198 superfamily. Furthermore, the existence of motif 16 leads to further functional differentiation. In conclusion, the functions of genes in the same group of the *DFR* gene family are similar, and the difference in the type and amount of conserved motif will lead to a change in gene function. 

### 2.4. Analysis of Expression Characteristics of DFR Gene Family

To further analyze the function of the *DFR* gene family, download the gene expression data of *A. thaliana* and *B. napus* from the Weigelworld database and the BrassicaEDB website. Flowers, leaves, roots, silique, and stems are selected to construct a heat map of expression data and combined with phylogenetic trees (Figure 6). It can be clearly seen from the figure that the expression of genes in Group I is basically non-expression or very low; the expression of *AtDFR3* in Group II is higher in flowers, and the expression of *BnDFRs* in silique is higher; and the expression of *AtDFR1* of Group III is higher in stems, the expression of *AtDFR4* in roots is higher, and the expression of *BnDFRs* is higher in flowers. The expression level of *AtDFRs* in Group IV is higher in silique, while the expression level of *BnDFRs* is higher in roots.

Cis-acting elements are transcription factor binding sites and other regulatory motifs with special functions in genes, which play an important role in the regulation of gene transcription initiation. The online website PlantCare is used to analyze the cis-acting elements within 2000 bp upstream of the *DFR* gene promoter. The cis-acting elements are drawn after the removal of two essential elements common in eukaryotes, CAAT-BOX and TATA-BOX, as well as some unnamed and uncommented elements (Figure 7). The remaining components contain a large number of elements related to hormone response and environmental stress response, such as abscisic acid responsiveness (ABRE); anaerobic induction (ARE); light-responsive elements, G-BOX, BOX 4, and MRE; and MeJA-responsive elements, CGTCA-motif, TGACG-motif, etc. There are 180 cis-acting elements in *A. thaliana*, among which the most elements are light-responsive elements at 38.33%, MeJA-responsive elements at 14.44%, anaerobic induction elements at 12.22%, and abscisic acid response elements at 10.56%. Likewise, there are 681 cis-acting elements in *B. napus*, of which the most are light-responsive elements accounting for 38.62%; MeJA-responsive elements accounting for 12.92%; abscisic acid response elements accounting for 10.43%; and anaerobic induction elements accounting for 10.43%. In addition, all 32 *DFR* genes contained light-responsive elements; only *BnDFR21* does not contain anaerobic induction elements. According to the analysis of cis-acting elements by different group members, it is found that there is no binding site of ATBP-1 element in Group I. The members of Group II do not contain abscisic acid responsiveness, flavonoid biosynthetic genes regulation, and MYBHv1 binding site elements. There is no flavonoid biosynthetic gene regulation and binding site of ATBP-1 in Group III, but the protein binding site element is unique in this group. Group IV contains the largest number of components, which basically contains all components related to anthocyanin biosynthesis in other groups. These results indicate that the expression of the *DFR* gene is regulated by various signals, and the functions performed by different groups of the *DFR* gene family are specific, suggesting that the *DFR* gene plays an important role in normal plant development.

### 2.5. Determination of Anthocyanin Content and Analysis of BnDFRs Expression in Rape of Different Color

In order to deeper understand the expression of the *DFR* gene in anthocyanin biosynthesis, solvent extraction is used to determine anthocyanin in the rape petals of five colors. According to the gene expression database, 11 genes with high expression in flowers are screened out for qRT-PCR to detect their expression levels (Figure 8). These 11 genes included 2 *BnDFRs* in Group I, 1 *BnDFR* in Group IV, and all *BnDFRs* in Group III. Additionally, the expression level of *BnDFRs* in Group III is significantly higher than that of other *BnDFRs*. As shown in Figure 8, the anthocyanin content of red, pink, and orange is significantly higher than that of yellow and white flowers, but only *BnDFR6* and *BnDFR26* in qRT-PCR results are consistent with this result, so it is speculated that *BnDFR6* and *BnDFR26* could promote anthocyanin biosynthesis and accumulation.

## 3. Discussion

*B. napus* is the main cultivated variety in China, which is planted in a large area in Hunan, Jiangxi, Anhui, and Zhejiang [26]. In rape festival tourism projects all over the country, colorful rape has been vigorously promoted and achieved remarkable results [27], indicating a broad market prospect and a high demand for colorful rape varieties, while the color of rape is the result of a combination of various factors. At present, research on the composition, biosynthetic pathway, and related genes of anthocyanin has been well understood [7]. With the development of transcriptomics and metabolomics, new research ideas have been provided for the analysis of the formation and regulation of the color of rape. For example, Yin et al. conducted a metabonomics analysis on the petals of four different colors of rape and determined the components and relative contents of flavonoids and anthocyanins in the petals. It was found that the content of anthocyanin in the petals of orange and pink flowers is significantly higher than that of yellow and white flowers, while there is no significant difference in the content of anthocyanin in yellow flowers and white flowers [28], which is consistent with the results of anthocyanin content measurement in this study.

*DFR* is the first key enzyme in the downstream pathway of anthocyanin biosynthesis, encoded by single or multiple genes. With the help of NADPH, and *ANS*, *UFGT*, it selectively catalyzed the formation of dihydroflavonol into pelargonidin, which made the color orange/brick red, catalyzed dihydroquercetin into cyanidin, which made the color magenta/red, and catalyzed dihydromyricone into delphiniside, which made the color blue/purple [29], plays a key role in anthocyanin biosynthesis. In recent years, a number of studies have shown that *DFR* is closely related to the flower color of many plants. For example, Fan Yan et al. studied two mutant 222-A-3 (near-white flower) and E30-D-1 (light-purple flower) and a near-isogenic line Clark-w4 (near-white flower) soybean. It is found that the *DFR* gene can regulate the formation of white to purple flowers of soybean [30]. Zhao et al. studied the expression of the *DFR* gene in different tissues of peonies and found that the highest expression is found in petals with high anthocyanin content [31], and similar reports were also reported in Asian lily [32] and gentian [33]. These results indicated that the *DFR* gene regulates flower color formation at the transcriptional level. In this study, qRT-PCR results also showed that the expression levels of *BnDFR6* and *BnDFR26* are consistent with the levels of anthocyanin content.

The model plant *A. thaliana* has obtained six *DFR* genes, which are responsible for different functions of anthocyanin biosynthesis. *AtDFR3* and *AtDFR5* play a role in the reduction in NADP-dependent flavonoids. *AtDFR1* and *AtDFR4* are responsible for specifically binding substrates and catalyzing specific enzymatic reactions; *AtDFR6* catalyzed the formation of colorless anthocyanins; *AtDFR2* is related to anthocyanin reduction reaction. The genome of *B. napus* has been sequenced, so we can systematically study and analyze the *DFR* family of *B. napus*. Six known *AtDFR* genes are used to identify, collate, and analyze the *DFR* gene family of *B. napus*. A total of 26 *BnDFRs* are identified and located on 17 chromosomes (9 in the A subgenome and 8 in the C subgenome) (Figure 2). Using the neighbor-joining (NJ) method, a phylogenetic tree was constructed by 26 members of *B. napus* and 6 members of *A. thaliana* (Figure 6). The phylogenetic results showed that except for three copies of *AtDFR3* and two copies of *AtDFR6* in *B. napus*, the remaining *AtDFRs* have four copies in *B. napus*, but in general, the gene in *A. thaliana* has six copies in *B. napus* [34]. In addition, interspecies collinearity analysis showed that there are three to six copies of *AtDFR* genes in the *B. napus* genome, indicating that the *DFR* gene had been lost during the formation and evolution of *B. napus*, and the remaining *BnDFR* genes may retain important functions. Although the sequence and molecular weight of *BnDFRs* vary greatly, the domain and motif composition are conserved.

In each group, most *BnDFRs* contain highly conserved motifs, and the types and amounts of motifs are different in each group (Figure 4). Therefore, we believe that the difference in motifs is the main cause of the functional differentiation of *BnDFRs*. In Group I, *BnDFR5* and *BnDFR11* apparently lost some conserved motifs in the evolutionary process, leading to functional changes. In Group III, functional changes are caused by the insertion of motif 18, and in Group IV, functional changes are also caused by the insertion of motif 17 and motif 20. Moreover, the addition of motif 16 resulted in further functional differentiation. Conserved functional domains are functional units of proteins that reveal the differentiated functions of different subfamilies, among which FR_SDR_e acts in the NADP-dependent reduction in flavonoids, ketone-containing plant secondary metabolites. The NADB_Rossmann superfamily is found in numerous dehydrogenases of metabolic pathways such as glycolysis and many other redox enzymes. PLN02214 is cinnamoyl-CoA reductase as a member of the NADB_Rossmann superfamily. PLN00198 superfamily is considered as anthocyanidin reductase provisionally, and PLN00198 is the only member of the PLN00198 superfamily. PLN02650 is dihydroflavonol-4-reductase. These results indicate that the functions of *BnDFRs* are differentiated, but they still play a certain role in anthocyanin synthesis.

Gene function is highly correlated with cis-acting elements in its upstream promoter region. Analysis of cis-acting elements in the promoter region of the *BnDFR* gene family showed that there are a large number of elements related to light response in all *BnDFR* genes (Figure 7), and it has been found that light can promote the expression of anthocyanin biosynthesis pathway structure genes and regulate genes. For example, strong light promoted the expression of structural genes *AtCHS*, *AtF3H*, *AtDFR*, and regulatory genes *AtPAP1* and *AtPAP2* for anthocyanin synthesis in *A. thaliana*, thus promoting the synthesis and accumulation of plant anthocyanins [35]. Strong light promoted the expression of *PhCHS*, *PhCHI*, and *PhFLS* in *Petunia*, while low light or darkness caused the expression of anthocyanin structural genes in *Petunia* and *Perilla* to be down-regulated or even not expressed, resulting in white or light-colored flowers [36,37]. The expression levels of structural genes such as *FaCHS*, *FaCHI*, *FaF3H*, *FaDFR*, *FaANS*, and *FaUFGT* and transcription factors *FaMYB10* and *FaMYB1* in strawberry decreased with the decrease in light intensity, and weak light weakened the red color of strawberry and reduced the anthocyanin content [38]. Tuberose is purplish red under strong light but will fade under weak light [39], which provides the basis for *BnDFR* genes to regulate color. There are other elements related to abiotic stress and hormone response, and there is specificity between different groups of elements. According to tissue-specific expression data, *BnDFRs* highly expressed in flowers are mainly in Group III, which contains protein binding site elements that are different from other groups. Such elements are mainly combined with transcription factors to regulate the expression of target genes [40]. At present, transcription factors affecting flower color include MYB, bHLH, and WD40 [41]. Therefore, it is speculated that *BnDFRs* containing protein binding site elements can bind transcription factors to promote the expression of the *DFR* gene, and different groups play different functions, which plays an important role in the growth and development of *B. napus*.

The subcellular localization of *DFR* proteins in *B. napus* is diverse and mainly exists in chloroplasts and the cytoplasm, among which all members of Group II exist in chloroplasts, and all members of Group III exist in the cytoplasm, as well as anthocyanins are synthesized in the cytoplasm and transferred to vacuoles after synthesis [42,43,44] (Table 1). Meanwhile, in tissue-specific expression data, the expression level of Group III in flowers is significantly higher than that of other groups, suggesting that members of Group III may play an important role in anthocyanin biosynthesis. Although studies have shown that *DFR* can catalyze three substrates in many plants, some *DFR*s can catalyze only one substrate with substrate specificity, such as in *Petunia* and *Cymbidium*, which cannot produce orange flowers based on pelargonium [45]. Johnson et al. [22] showed that the substrate specificity of *Gerbera* was directly determined by the amino acid residue between positions 134 and 145. According to the amino acid residues at the 134th position, *DFR* is divided into 3 categories. The first category is aspartamide (Asn)-type *DFR*. Most plant *DFR* is of this type, and its amino acid residues are Asn. The second type is aspartic acid (Asp)-type *DFR*, whose amino acid residue is Asp, which has no catalytic activity for DHK, such as *Petunia*. The third class is non-Asn/Asp-type *DFR*s, whose amino acid residues are neither Asn nor Asp. Among the 26 *BnDFR* in this study, *BnDFR6*, *BnDFR13*, *BnDFR20*, and *BnDFR25* are Asp-type *DFR*s, and all belong to Group III. The remaining *DFR*s are non-Asn/Asp-type *DFR*s. In addition, some *BnDFR* was highly expressed in materials with low anthocyanin content, and others were highly expressed in materials with high anthocyanin content, suggesting that *BnDFRs* may have substrate specificity (Figure 8). Whether the difference in flower color is affected by the *DFR* effect of a single gene or the result of the superposition of multiple genes, as well as how to regulate and coordinate the effect, still need to be further studied. The *DFR* genes play an important role in the development and coloration of *B. napus*. The exact role of *BnDFR* gene family members in *B. napus* has not been clarified. In this study, 26 *BnDFR* family members in *B. napus* are identified through bioinformatics analysis. The *BnDFR* genes are systematically studied from the aspects of evolution, gene location, gene structure, protein physicochemical properties, expression characteristics, and so on, the tissue specificity of the *BnDFRs* gene was revealed, and the correlation between the expression of *BnDFRs* and the synthesis of anthocyanins in rape petal was analyzed. Learned that the expression levels of *BnDFR6* and *BnDFR26* are consistent with the content of anthocyanins, so it is speculated that *BnDFR6* and *BnDFR26* play an important role in the synthesis and accumulation of anthocyanins in rape petals. However, the specific function and response mechanism still need further study. This study provides a theoretical basis and bioinformatics reference for the function of the *DFR* gene and its involvement in anthocyanin biosynthesis in *B. napus.* Moreover, it will be helpful for further research on the evolutionary origin of *BnDFRs* and the function of *BnDFRs* candidate genes in rape molecular breeding.

## 4. Materials and Methods

### 4.1. Materials and Data Sources

Rape of different colors is planted in the campus experimental field (31.92° N, 117.21° E). The *A. thaliana* and *B. napus* protein files and associated annotation files can be downloaded from the Ensembl Plants website (http://plants.ensembl.org/index.html (accessed on 23 November 2022)).

### 4.2. Analysis of BnDFR Family Proteins

Search and download the dihydroflavonol 4-reductase subfamily protein sequence at UniProt website (https://www.uniprot.org/ (accessed on 23 November 2022)) [46], then BlastP is performed on Ensembl Plants website with *B. napus* genome database using *Arabidopsis DFR* gene family, and *BnDFR* candidate genes are selected according to E-value < 1 × 10^−50^.

Using the ProtParam tool (https://web.expasy.org/protparam/ (accessed on 27 November 2022)), we analyzed *A. thaliana* and *B. napus DFR* proteins [47] of the physicochemical properties. It includes molecular weight (MW), theoretical point of isoelectric (pI), instability index (II), and grand average of hydrophobicity (GRAVY). Moreover, using the Wolf Psort (https://www.genscript.com/wolf-psort.html (accessed on 9 December 2022)) predicts the *DFR* protein subcellular localization (SL) [48]. The protein homology/analogy recognition engine website PHYRE2 was used to predict *BnDFRs* 3D protein structure [49]. The α-helices, β-turns, and other information are analyzed by the protein secondary structure prediction website SOPMA; this website is easy to use and has a simple interface that is used in most articles.

### 4.3. BnDFR Gene Location and Collinearity Analysis

Download the DNA file and annotation file (GTF) of *A. thaliana* and *B. napus* on Ensembl Plants website. The interspecies collinearity and intraspecies collinearity analysis diagrams are drawn by using One Step MCScanX and Advance Circos functions of TBtools [50]. Use Gene Location Visualize from GTF/GFF function to map the distribution of the *BnDFR* gene family on chromosomes.

### 4.4. Analysis of Gene Structure, Conserved Motifs, and Conserved Domain of DFR Family Proteins

After downloading the gene structure annotation file (GTF file) on Ensembl Plants website. The structure information of the *DFR* Gene family in *A. thaliana* and *B. napus* is visualized by using Gene Structure View (Advanced) function in TBtools software. The conservative analysis of *DFR* protein sequences is implemented by MEME (https://meme-suite.org/meme/tools/meme (accessed on 4 February 2023)) [51] and the Simple MEME Wrapper function of the TBtools software, set 20 Motifs and the number of motif occurrences on each sequence is not limited. Other parameters are default. Conservative structure domain is analyzed by the CD search function of NCBI (https://www.ncbi.nlm.nih.gov/Structure/cdd/wrpsb.cgi (accessed on 21 February 2023)) [52,53].

### 4.5. Cis-Acting Element Analysis of DFR Gene Promoter Region

Gtf/Gff3 Sequences Extract in TBtools is used to extract the upstream 2000 bp parameters of CDS and extract the promoter sequence of *DFR* genes, submitted to PlantCARE database (http://bioinformatics.psb.ugent.be/webtools/plantcare/html/ (accessed on 26 November 2022)) to analyze its promoter region cis-acting element [54], and the resulting file is filtered based on the information in the table and retained for viewing purposes, then visually presented by using the Simple BioSequence Viewer function of TBtools software.

### 4.6. Evolutionary Analysis and Tissue Expression Specificity Analysis of DFR Gene Family

MEGA11 software is used to construct the phylogenetic tree of *DFR* genes of *A. thaliana* and *B. napus*. Match the full-length sequences of *DFR* proteins with each other, and the phylogenetic analysis is performed. The neighbor-joining (NJ) tree is constructed with parameters set to bootstrap = 1000, then using the iTOL (https://itol.embl.de/ (accessed on 14 April 2023)), an online mapping website, to beautify the evolutionary tree [55]. The expression data of *A. thaliana* and *B. napus* are downloaded from Weigelworld database (https://weigelworld.org/resources.html (accessed on 18 February 2023)) and BrassicaEDB (https://brassica.biodb.org/ (accessed on 18 February 2023)). Five tissues, root, stem, leaf, flower, and silique, are selected [56]. The gene expression data is imported into iTOL website and combined with *DFR* gene family phylogenetic tree for display.

### 4.7. Anthocyanin Content Determination, Expression Modeling Analysis of the BnDFRs, and qRT-PCR Analysis

The petals of different color rape are respectively added to 1 mL 1% HCl methanol solution to extract anthocyanin for 18 h and then centrifuged (14,000 r/min). After that, add 0.4 mL supernatant to 0.6 mL 1% HCl methanol solution. Then, measure the light absorption values at 530 nm and 657 nm wavelengths by spectrophotometer and calculate the anthocyanin content according to the formula below [57]. Three biological replicates are taken for each material, and the average value is calculated at last.
Q_anthocyanins_ = (OD_A530_ − 0.25 × OD_A657_)/M(1)
Note: Q_anthocyanins_ is the anthocyanins content, ODA530 and ODA657 are absorbance values at 530 nm and 657 nm wavelengths, respectively, and M is the sample mass.

The petals of the same plants with anthocyanin content are taken, and the total RNA of the material is extracted using TaKaRa MiniBEST Plant RNA Extraction Kit, and cDNA is synthesized using PrimeScript^TM^ RT reagent Kit with gDNA Eraser (Perfect Real Time), both are manufactured by Takaeabio company. The qRT-PCR specific primers are designed by Primer Premier 5 (Table 3), with BnActin chosen as the internal genes; set up three biology repetitive and three technology, using 2^−ΔΔCt^ methods to analyze gene expression [58].

## Figures and Tables

**Figure 1 plants-12-02583-f001:**
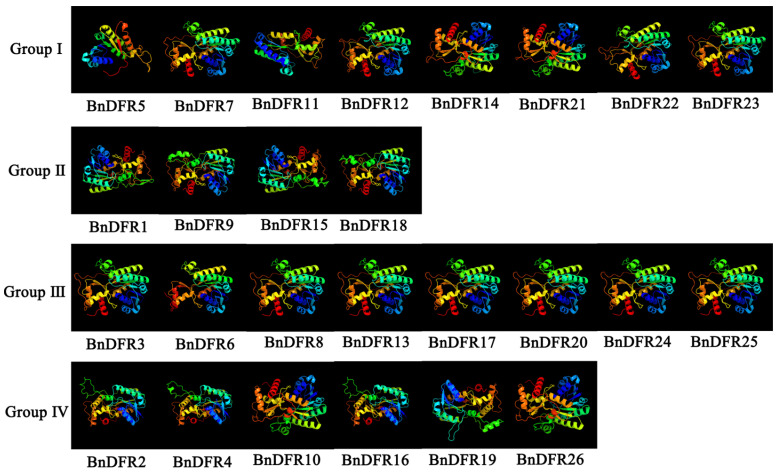
Three-dimensional structures of the *BnDFR* proteins.

**Figure 2 plants-12-02583-f002:**
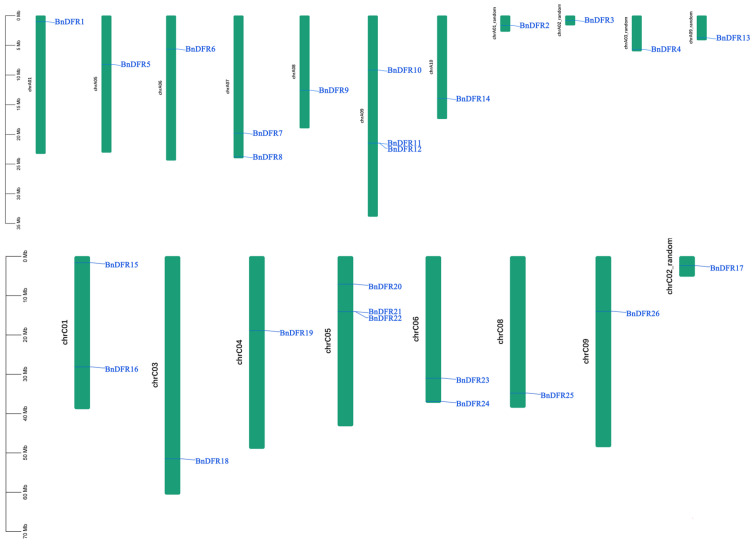
*DFR* genes distribution on *B. napus* chromosomes.

**Figure 3 plants-12-02583-f003:**
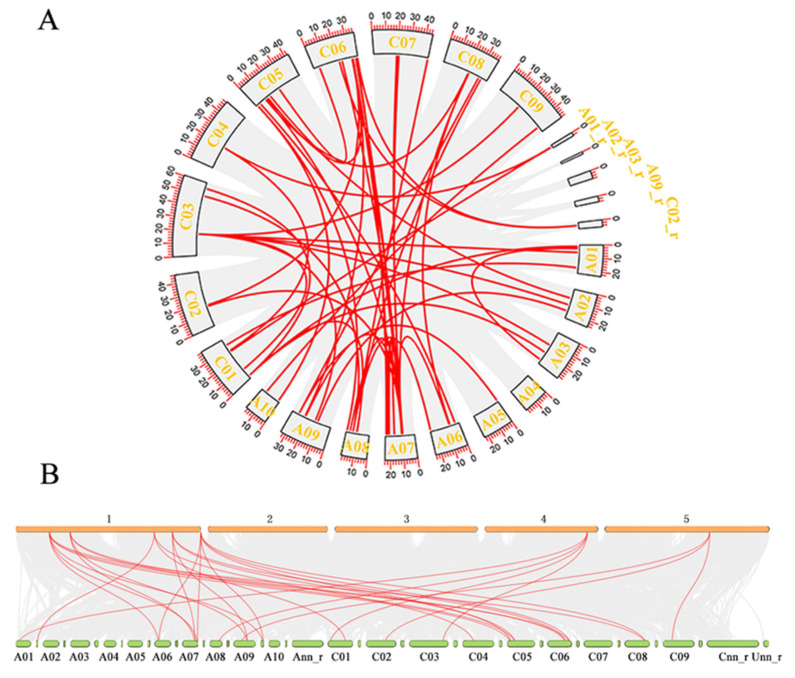
Collinearity analysis: intraspecies collinearity analysis (**A**) and interspecies collinearity analysis (**B**). Note: The gray background line represents the collinear block in the whole genome, and the highlighted line represents the collinear relationship. In subfigure B, the top 1-5 is the chromosome name of *A. thaliana*, the bottom A01-Unn_r is the chromosome name of *B. napus*, and the small chromosome fragment on the right of each chromosome is its corresponding random.

**Figure 4 plants-12-02583-f004:**
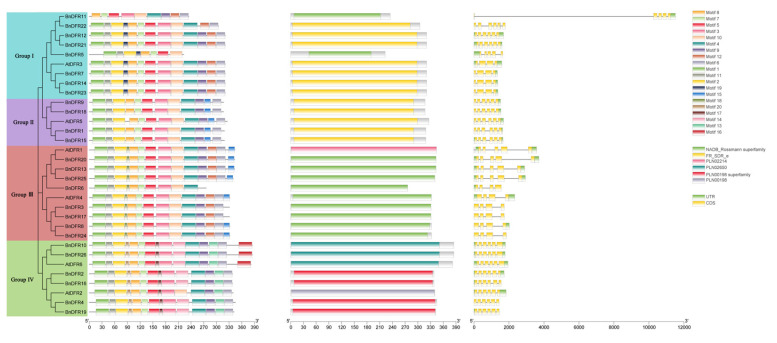
Phylogenetic tree, motif, domain, and gene structures of *DFR* gene family.

**Figure 5 plants-12-02583-f005:**
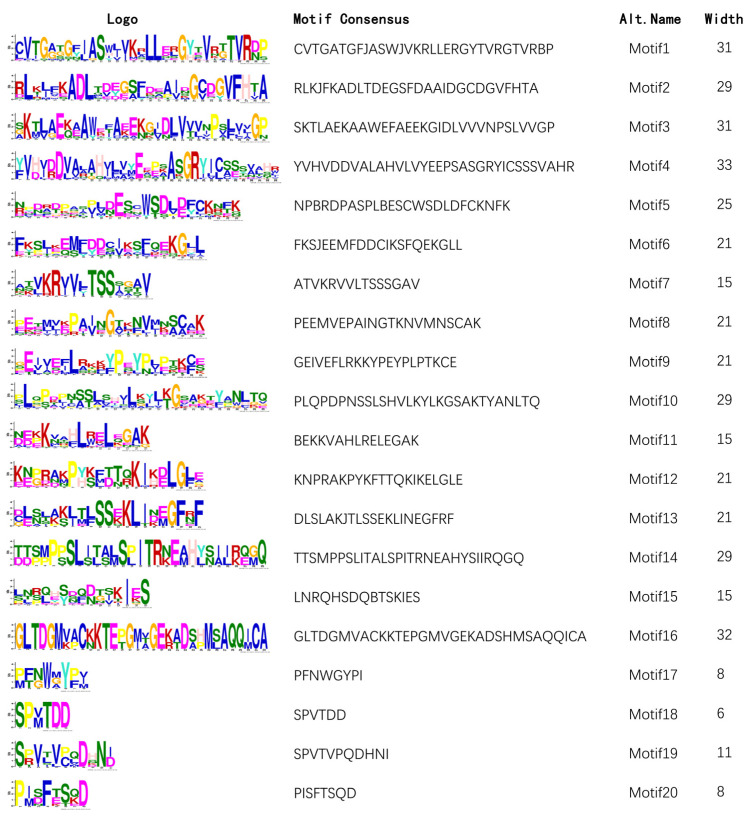
Highly conserved motif patterns of *DFR* gene family.

**Figure 6 plants-12-02583-f006:**
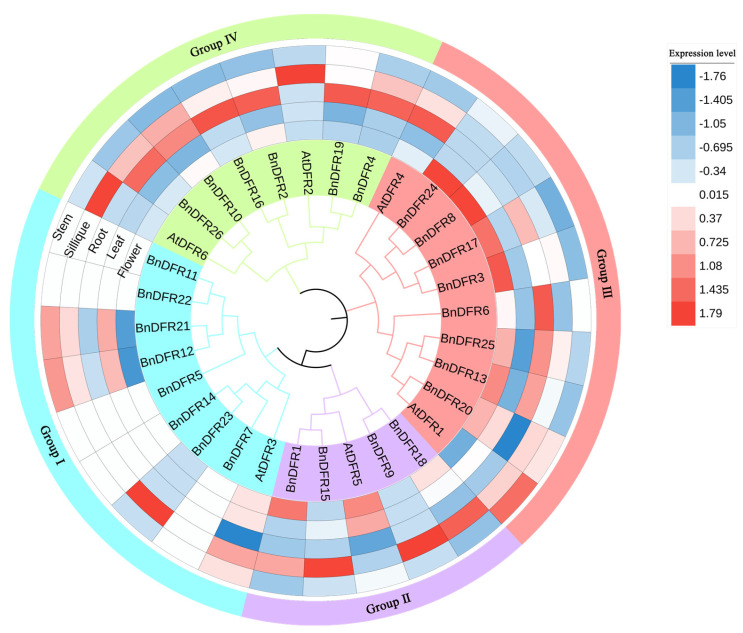
Phylogenetic tree and specific expression of *DFR* gene family.

**Figure 7 plants-12-02583-f007:**
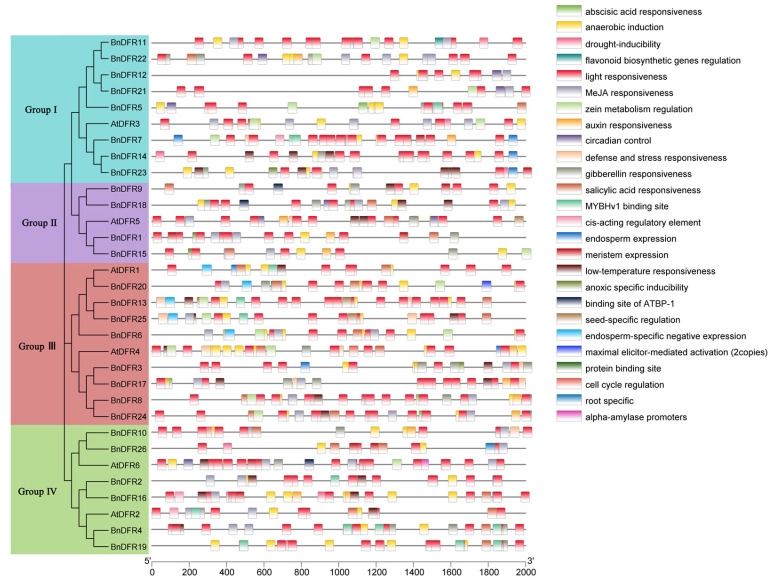
Cis-acting element structures in promoter regions of *DFR* family members.

**Figure 8 plants-12-02583-f008:**
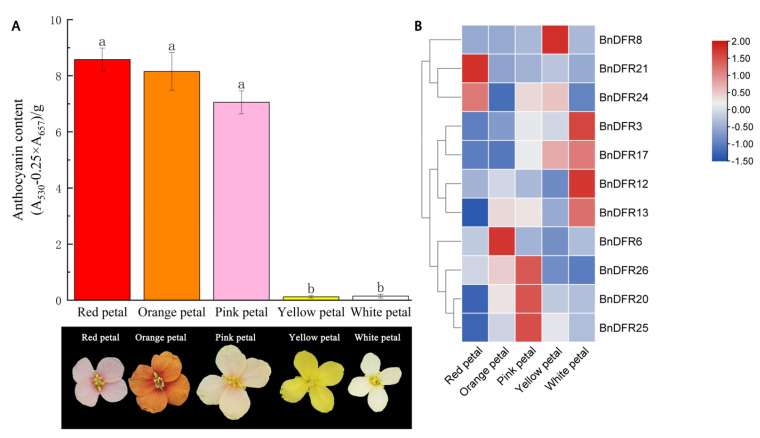
Anthocyanin content (**A**) and qRT-PCR analysis (**B**) of some *BnDFR* genes in *B. napus*. (**A**) shows different color rape and its anthocyanin content, (**B**) shows the gene expression levels of some *BnDFRs* and normalize it with a row scale.

**Table 1 plants-12-02583-t001:** Physicochemical property and subcellular localization of *DFR* proteins.

Gene	Len (aa)	MW (kD)	pI	II	GR	SL
*AtDFR1*	344	37.49	6.13	32.17	−0.26	Cytoplasm
*AtDFR2*	340	37.91	5.89	26.2	−0.176	Nucleus
*AtDFR3*	321	35.67	6.02	39.55	−0.159	Nucleus
*AtDFR4*	332	36.62	6.4	25.89	−0.166	Cytoplasm
*AtDFR5*	326	36.46	5.88	38.22	−0.122	Chloroplast
*AtDFR6*	382	42.77	5.43	36.85	−0.225	Chloroplast
*BnDFR1*	319	35.74	5.95	36.76	−0.093	Chloroplast
*BnDFR2*	338	37.7	5.07	37.27	−0.203	Cytoplasm
*BnDFR3*	331	36.51	6.71	30.04	−0.193	Cytoplasm
*BnDFR4*	345	38.26	5.96	34.92	−0.171	Nucleus
*BnDFR5*	223	25.08	8.76	41.34	−0.526	Mitochondrion
*BnDFR6*	276	30.04	5.35	28.51	−0.048	Cytoplasm
*BnDFR7*	321	35.68	6.52	36.34	−0.168	Nucleus
*BnDFR8*	332	36.49	6.19	30.6	−0.195	Cytoplasm
*BnDFR9*	317	35.54	6.27	35.05	−0.14	Chloroplast
*BnDFR10*	385	42.89	5.54	37.87	−0.255	Chloroplast
*BnDFR11*	235	26.61	8.79	38.4	−0.206	Nucleus
*BnDFR12*	321	35.82	5.5	37.27	−0.172	Chloroplast
*BnDFR13*	343	37.51	5.53	33.85	−0.271	Cytoplasm
*BnDFR14*	321	35.73	6.52	37.77	−0.162	Nucleus
*BnDFR15*	319	35.62	5.96	36.23	−0.127	Chloroplast
*BnDFR16*	338	37.68	4.93	37.03	−0.164	Cytoplasm
*BnDFR17*	331	36.42	6.66	32.36	−0.182	Cytoplasm
*BnDFR18*	317	35.46	5.55	43.06	−0.138	Chloroplast
*BnDFR19*	342	37.87	5.8	34.06	−0.156	Nucleus
*BnDFR20*	343	37.56	5.79	34.7	−0.29	Cytoplasm
*BnDFR21*	321	35.85	5.5	36.64	−0.171	Chloroplast
*BnDFR22*	305	34.1	5.47	34.04	−0.125	Cytoplasm
*BnDFR23*	321	35.79	6.52	37.84	−0.219	Nucleus
*BnDFR24*	332	36.57	6.26	31.35	−0.202	Cytoplasm
*BnDFR25*	340	37.22	5.53	32.91	−0.254	Cytoplasm
*BnDFR26*	385	42.93	5.64	37.07	−0.256	Chloroplast

Note: Len—protein sequence length; MW—molecular weight; pI—theoretical isoelectric point; II—instability index, unstable proteins are marked in red; GR—grand average of hydrophobicity; SL—subcellular localization.

**Table 2 plants-12-02583-t002:** Results and validity of secondary structures and 3D structures of *BnDFR* proteins.

	Secondary Structures				3D Structures	
Gene	α-Helix	Extended Strand	β-Turn	Random Coil	Confidence	Coverage
*BnDFR1*	40.44	15.32	7.52	36.68	100.0%	95.0%
*BnDFR2*	42.9	14.79	7.4	34.91	100.0%	92.0%
*BnDFR3*	44.11	15.41	6.65	33.84	100.0%	96.0%
*BnDFR4*	39.71	15.94	7.25	37.1	100.0%	90.0%
*BnDFR5*	46.19	12.11	4.93	36.77	99.8%	81.0%
*BnDFR6*	48.19	17.03	5.43	29.35	100.0%	97.0%
*BnDFR7*	42.06	13.08	5.92	38.94	100.0%	97.0%
*BnDFR8*	45.78	15.36	5.72	33.13	100.0%	96.0%
*BnDFR9*	43.22	15.77	7.89	33.12	100.0%	96.0%
*BnDFR10*	40.52	11.95	7.53	40	100.0%	83.0%
*BnDFR11*	40.85	12.77	5.11	41.28	99.9%	97.0%
*BnDFR12*	41.74	14.95	7.17	36.14	100.0%	97.0%
*BnDFR13*	42.27	14.29	5.25	38.19	100.0%	92.0%
*BnDFR14*	39.88	14.64	6.54	38.94	100.0%	95.0%
*BnDFR15*	42.63	14.73	7.84	34.8	100.0%	95.0%
*BnDFR16*	42.6	14.2	7.1	36.09	100.0%	92.0%
*BnDFR17*	45.92	15.71	5.44	32.93	100.0%	96.0%
*BnDFR18*	41.01	16.72	7.89	34.38	100.0%	96.0%
*BnDFR19*	41.52	14.33	7.31	36.84	100.0%	91.0%
*BnDFR20*	43.73	13.12	5.25	37.9	100.0%	92.0%
*BnDFR21*	41.74	15.26	5.92	39.07	100.0%	97.0%
*BnDFR22*	41.31	16.07	7.21	35.41	100.0%	97.0%
*BnDFR23*	43.3	14.95	6.85	34.89	100.0%	97.0%
*BnDFR24*	45.78	15.96	6.02	32.23	100.0%	95.0%
*BnDFR25*	45	12.94	5	37.06	100.0%	94.0%
*BnDFR26*	42.6	13.25	6.23	37.92	100.0%	83.0%

**Table 3 plants-12-02583-t003:** Primer sequence.

Gene Name	Forward Primer Sequence	Reverse Primer Sequence
*BnActin7*	AACCTTCTCTCAAGTCTCTGTG	CCAGAATCATCACAAAGCATCC
*BnDFR3*	GTGCTGATCTTCTCGACTATGA	TCACGGTTAGGGTTCATGTAAA
*BnDFR6*	ATATACTTTCGACCGCTGGAAA	TCCAAGAAGCTATGTATCCACC
*BnDFR8*	CGATAAGAATCCAAGGGCAAAG	TGATGGTCGTATCGTGTTCTAG
*BnDFR12*	CATTTGAGACCAAGTGCAGTAG	GCCAAGTTCACGTATCTTTGTT
*BnDFR13*	AATTGGTGCAGTTTACATGGAC	GGTGTTTTTGCAGAACTCAAGA
*BnDFR17*	AGTATCCGCTTCCTATCAAGTG	GCTCTTGACAGATTCGTAGAGA
*BnDFR20*	GTTAAGAGCTTGCAAGAGAAGG	ATCGCTGAGGATGATACTTCAG
*BnDFR21*	GCTAAAGATTTTCGAAGCCGAT	TTCTCCAGATCATTGTTGTCCA
*BnDFR24*	TTTACATGAACCCTAACCGTCA	TAAGGTTAGCATAGGTCTTGGC
*BnDFR25*	CAGGACTACGATGCTCTTAAGT	AATTACAAACTTGGCTCCGTTC
*BnDFR26*	CTGCCAAGGGACGTTATATTTG	CAAACGTTGAAGGCACGTTATA

## Data Availability

Not applicable.

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
