# Peer review of "Identification and Expression Analysis of DFR Gene Family in Brassica napus L."

_plants, 2023, doi:10.3390/plants12132583_

Round 1

Reviewer 1 Report

This article presented Identification and Expression Analysis of DFR Gene Family in Brassica napus L. This provides important theoretical guidance for further analysis of anthocyanin anabolism mechanism involved in DFR gene in Brassica napus L. Before recommending this article for publication, there are some shortcomings for that should be resolve.

The abstract of the article did not provide brief information of the study. Specifically, results are missing in the results.

First sentence of the abstract is too long and void in the meaning. It should be revised.

Italicize the species name in all the MS.

Add information of the origin and genetic basis of the B. napus

Line 39 should be cited with recent studies https://doi.org/10.1016/j.foodres.2023.112637, https://doi.org/10.1016/j.indcrop.2022.116090

Line 40-43 the reference style must be according to journal guidelines.

Second paragraph of the introduction must add how genetic studies such as genome wide identification can be helpful in identifying color specific genes.

Line 68-69 taxa names must be italic.

Use B. napus after first use of the Brassica napus.

The quality of figure 4 is very low.

Discussion section should explain and compare the results.

Section 4.4 conserved motifs provide complete details and should be cited with a recent study https://doi.org/10.1016/j.plaphy.2021.01.042

The conclusion line 306 to 312 need to be revise by presenting key findings in specific terms and future directions in general based on the obtained results.

For English I would recommend to avoid long sentences and italicize all species names and also check consistency of the abbreviations

references must be according to journal guidelines 

Reviewer 2 Report

This manuscript, which comprehensively examines the DFR gene in rape, is expected to provide important fundamental knowledge for understanding anthocyanin synthesis and its diversity in rape and related plant species. However, there were many missing or inaccurate explanations and descriptions, and the order of composition should also be considered.

Major comments

1) The figures should be described so the reader can learn all the information in the figure with only the figure and figure description. The description of the tools used, the legend, and the elements in the figures should be sufficiently described.

2) 3D structure of BnDFRs should be shown in groups' separation in Figures 4 and 6. Please reconsider the order of description and discussion.

3) There are other secondary structure prediction tools, such as AlphaFold2. If there is a reason for selecting tools in this study, it would be better to describe it briefly.

4) There is not enough information for the reader to evaluate the validity of the results regarding the structure prediction.

It is stated that the analysis was performed using Phyre2 and SOPMA, but no description allows the reader to determine which parameters were obtained with which tool.

5) It would be better to describe not only the 3D model by Phyre2 but also the numerical values to evaluate the validity of the results. Also, it would be better to clearly state the results obtained by SOPMA in a table or other format.

6) Some DFRs, such as the DFRs of roses, show substrate selectivity in the anthocyanidin synthesis process, and it would be helpful to add analysis and discussion of whether any of the DFRs of rapes have such selectivity, if possible.

Minor comments

7) L26 Brassica napus L Brassica napus L. or Brassica napus

8) L83 Brassica napus should be in italics. Please check all Brassica napus.

9) L83 napus,Blast napus, Blast

10) L84 Arabidopsis thaliana should be in italics. Please check all.

11) L96 Citation and URL are necessary for PHYRE2 in the first appearance.

12) L98 Citation and URL are necessary for SOPMA in the first appearance.

13) L154-155 motif15 motif 15

14) L279 in different groups are different are different in each group  ?

15) L379 Takara RNA extraction kit Please describe the correct name.

16) L382 Which company made this kit?

17) L382 There is no information about the concentration of primers and templates. Authors can describe only which kit was used.

18) GroupI~IV should be described by I, II, III and IV, with no environment-dependent characters.

19) How to normalize the expression value in Figure 8 ?

20) In each group, most BnDFRs contain highly conserved motifs, and the types and amounts of motifs in different groups are different.

Since there were few comments, they are listed in Comments and Suggestions for Authors.
